# Constrained Reinforcement Learning with Smoothed Log Barrier Function

**Baohe Zhang**                                                     *zhangb@cs.uni-freiburg.de*
*Department of Computer Science*
*University of Freiburg*

**Yuan Zhang**                                                      *yzhang@cs.uni-freiburg.de*
*Department of Computer Science*
*University of Freiburg*

**Hao Zhu**                                                         *zhuh@cs.uni-freiburg.de*
*Department of Computer Science*
*University of Freiburg*

**Shengchao Yan**                                                   *yan@cs.uni-freiburg.de*
*Department of Computer Science*
*University of Freiburg*

**Thomas Brox**                                                     *brox@cs.uni-freiburg.de*
*Department of Computer Science*
*University of Freiburg*

**Joschka Boedecker**                                              *jboedeck@cs.uni-freiburg.de*
*Department of Computer Science*
*University of Freiburg*

**Reviewed on OpenReview:** *https://openreview.net/forum?id=Amh95oURaE*

## Abstract

Deploying reinforcement learning (RL) in real-world systems often requires satisfying strict safety constraints during both training and deployment, which simple reward shaping typically fails to enforce. Existing constrained RL algorithms frequently face several major challenges, including instabilities during training and overly conservative policies. To overcome these limitations, we propose CSAC-LB (**C**onstrained **S**oft **A**ctor-**C**ritic with **L**og **B**arrier), a model-free, sample-efficient, off-policy algorithm that requires no pre-training. CSAC-LB integrates a linear smoothed log barrier function into the actor's objective, providing a numerically stable, non-vanishing gradient that enables the agent to quickly recover from unsafe states while avoiding the instability of traditional interior-point methods. To further enhance safety and mitigate the underestimation of constraint violations, we employ a pessimistic double-critic architecture for the cost function, taking the maximum of two cost Q-networks to conservatively guide the policy. Through extensive experiments on challenging constrained control tasks, we demonstrate that CSAC-LB significantly outperforms baselines by consistently achieving high returns while strictly adhering to safety constraints. Our results establish CSAC-LB as a robust and stable solution for applying RL to safety-critical domains.

# 1    Introduction

In traditional Reinforcement Learning (RL) (Sutton & Barto, 1998) formulations, rewards are usually the only metric of performance. However, in many real-world applications, the performance of an algorithm is not necessarily measured by a single objective. For example, in the context of an autonomous driving task, the agent must drive as quickly as possible while still adhering to traffic rules and avoiding collisions. Designing a reward function that considers all individual cases can be challenging. Reward engineering, which involves tuning the weights of different reward components, may also lead to a suboptimal solution. In the context of deep reinforcement learning, where neural networks are used as function approximators, it is often not feasible to apply conventional constrained optimization methods due to the large number of parameters to be optimized.

If no prior knowledge is available, constraint violation is unavoidable during training as the RL agent must visit the unsafe states of the environment to gradually learn a safe policy. A fundamental challenge is to predict the boundary of the safe region using a neural network, since neural networks typically cannot extrapolate well to unseen data (Barnard & Wessels, 1992). To predict this safe margin, the agent must first see some samples from the unfeasible set. However, dramatically violating constraints is also generally undesirable, as this may cause severe damage to the system being optimized. This means an RL agent must avoid extensive constraint violations and learn efficiently from as few violations as possible.

Considering these challenges, existing approaches often require limiting assumptions, such as a known suboptimal recovery policy (Thananjeyan et al., 2021; Banerjee et al., 2024) or prior knowledge of the system dynamics (Liu et al., 2025; Luo & Ma, 2021). This restricts generalization and requires domain expertise. The challenge is particularly critical in tasks where the optimal policy lies on the boundary of the safe region. Success in these tasks hinges directly on the agent's ability to explore the safe margin efficiently. This requirement, however, reveals a common weakness in existing algorithms (Achiam, 2021) as found in Marchesini et al. (2022), which often fail to strike the necessary balance, resulting in policies that are either too reckless and unsafe or too conservative and suboptimal.

Motivated by the above, we propose CSAC-LB, which applies a linear smoothed log barrier function (Kervadec et al., 2019) to the Soft Actor-Critic (SAC) algorithm (Haarnoja et al., 2018) augmented by a safety critic (Srinivasan et al., 2020). It serves as a general-purpose method that can handle various types of cost functions without requiring any additional information or pre-training. Being an off-policy algorithm, it can leverage data collected previously for efficient learning. The goal of this algorithm is to effectively explore the safe margin of a given problem during training and to learn a well-performing policy for future deployment in real-world tasks. Crucially, the smoothed log barrier provides a stable, non-vanishing gradient even in unsafe states, enabling the agent to recover quickly from constraint violations and to explore the boundary between safe and unsafe regions without suffering from numerical instability or vanishing gradients. This facilitates more effective and targeted exploration, allowing the agent to learn the true constraint boundary with fewer violations and greater sample efficiency than prior methods. This distinguishes it from other works that aim for zero constraint violations during training.

Our contributions are the following:

- We propose a new actor objective that integrates a linear smoothed log barrier function. This function provides a stable, non-vanishing gradient signal even when the policy is in an unsafe state, allowing for rapid recovery without the numerical issues of standard log barriers or the gradient saturation of clipping methods. This enables effective exploration along the constraint margin, which is critical for learning in tasks where optimal policies lie near the feasibility boundary.

- To combat the critical problem of cost underestimation by the critic network, we employ a pessimistic double-critic architecture for the cost function. By taking the maximum of the two cost Q-values, we ensure a conservative estimate of constraint violations, leading to more reliable and safer policies.

- We provide a theoretical analysis of CSAC-LB, establishing a bound on its suboptimality and guaranteeing bounded constraint violation. Through extensive experiments on ten challenging benchmark

tasks, we demonstrate that CSAC-LB consistently outperforms state-of-the-art baselines, achieving a superior balance of high returns, strict constraint satisfaction, and stable convergence.

## 2 Related Work

Many prior works have been proposed for addressing the safety of reinforcement learning using various approaches. The reviews by García & Fernández (2015), Dulac-Arnold et al. (2021), and Ma et al. (2022) provide a comprehensive overview.

One direction which has been explored for solving constrained RL problems is safe policy search. Constrained Policy Optimization (CPO) (Achiam et al., 2017) is the first general-purpose approach that solves constrained reinforcement learning using a trust-region method and is able to provide certain theoretical guarantees. Polymenakos et al. (2019) extend safe policy search by augmenting it with a Gaussian Process model for risk estimation. Other approaches leverage cost or safety critics to guide the policy. For instance, Ha et al. (2020) extend SAC (Haarnoja et al., 2018) with a cost function and optimize the constrained problem by introducing a Lagrange-multiplier. However, this method suffers in terms of training robustness when constraint violations occur only rarely during training on the task. Building on this idea, the method by Srinivasan et al. (2020) learns to predict the cumulative constraint violations and update the policy accordingly. Further improvements are presented by Yang et al. (2021) where a distributional safety critic is combined with the CVaR metric. By changing the percentile of this CVaR metric, the sensitivity to risk can also be changed accordingly. The work presented by Ying et al. (2022) also uses CVaR as the metric, but in conjunction with the on-policy method Proximal Policy Optimization (PPO) (Schulman et al., 2017) as the RL algorithm. In a separate line of inquiry, Lyapunov functions are also widely applied to constrained RL problems (Chow et al., 2018; 2019). By projecting policy parameters onto the feasible solutions from linearized Lyapunov constraints during the policy update, these methods can be applied to any policy gradient method, such as DDPG (Lillicrap et al., 2016) or PPO.

Another line of work is close to Model Predictive Control (MPC) (Rawlings et al., 2017). Model-based approaches, such as (Berkenkamp et al., 2017; Cheng et al., 2019), use a dynamics model to certify the safety of the system. These approaches usually assume that the dynamics model is calibrated and the constraints are known. This, however, limits the general applicability of the algorithms. A framework for learning barrier certificates (Ames et al., 2019) and the policy iteratively, achieving zero training-time constraint violations in an empirical analysis, is presented by Luo & Ma (2021). The work by Pereira et al. (2020) combines stochastic barrier functions with safe trajectory optimization and is able to recover the optimal policy under certain conditions. As et al. (2022) improve data efficiency in constrained RL by using a Bayesian World model. Huang et al. (2024) learns a world model for training and planning, leading to a higher data-efficiency.

Close to our work is the application of interior-point methods to PPO (Zeng & Zhang, 2018; Liu et al., 2020). However, these methods are on-policy methods and have numerical stability issues due to the log barrier function. To the best of our knowledge, we are the first to propose a numerically stable, off-policy interior-point algorithm. By integrating a novel linear smoothed log barrier with the highly sample-efficient SAC framework, our work, CSAC-LB, addresses the critical challenges of both training instability and sample efficiency in constrained reinforcement learning.

## 3 Background

### 3.1 Constrained Markov Decision Processes

A Markov Decision Process (MDP) provides the mathematical foundation for sequential decision-making. It is formally defined as a tuple $(\mathcal{S}, \mathcal{A}, P, R, \gamma)$, where $\mathcal{S}$ is the state space, $\mathcal{A}$ is the action space, $P: \mathcal{S} \times \mathcal{A} \times \mathcal{S} \to [0,1]$ is the state transition probability function, $R: \mathcal{S} \times \mathcal{A} \to \mathbb{R}$ is the reward function and $\gamma \in [0,1)$ is the discount factor.

An agent interacts with the environment by following a policy $\pi: \mathcal{S} \to \mathcal{P}(\mathcal{A})$, which maps states to a probability distribution over actions. The primary goal in reinforcement learning is to find an optimal policy

where the expected cumulative discounted reward, given by

$$J_R(\pi) = \mathop{\mathbb{E}}_{(s_t,a_t)\sim\pi}\left[\sum_{t=0}^{\infty}\gamma^t R(s_t,a_t)\right], \tag{3.1}$$

is maximized.

A Constrained Markov Decision Process (CMDP) (Eitan, 1999) extends the MDP framework by augmenting the MDP tuple with a set of cost functions, $\{C_1, \ldots, C_m\}$, where $C_i\colon \mathcal{S} \times \mathcal{A} \to \mathbb{R}$, $i = 1, \ldots, m$, maps a given state-action pair to the constraint violation penalty. For each cost function, we define an expected cumulative discounted cost as

$$J_{C_i}(\pi) = \mathop{\mathbb{E}}_{(s_t,a_t)\sim\pi}\left[\sum_{t=0}^{\infty}\gamma^t C_i(s_t,a_t)\right]. \tag{3.2}$$

The objective in a CMDP is to find a policy that maximizes the expected return (3.1) while ensuring that each expected cost (3.2) remains below a predefined threshold $d_i$. Let $\Pi$ denote the set of all possible policies, then this leads to the following constrained optimization problem:

$$\begin{aligned}
&\underset{\pi\in\Pi}{\text{maximize}} && J_R(\pi)\\
&\text{subject to} && J_{C_i}(\pi) \leq d_i, \quad i = 1, \ldots, m.
\end{aligned} \tag{3.3}$$

## 3.2 SAC-Lagrangian

**The Lagrange Multiplier Method** The Lagrange multiplier method is a standard technique for solving constrained optimization problems (Bertsekas, 1982). Let $f, g\colon \mathbb{R}^n \to \mathbb{R}$ be real valued functions on $\mathbb{R}^n$, a general constrained optimization problem, often called the primal problem, can be written as:

$$\begin{aligned}
&\underset{x\in\mathbb{R}^n}{\text{maximize}} && f(x)\\
&\text{subject to} && g(x) \leq 0.
\end{aligned} \tag{3.4}$$

This problem can be transformed into an unconstrained optimization by introducing the Lagrangian function:

$$L(x,\lambda) = f(x) - \lambda g(x),$$

where $\lambda \geq 0$ is the Lagrange multiplier. The original problem (3.4) is then transformed to solving the following saddle-point (or max-min) problem, known as the Lagrangian dual problem:

$$\underset{x\in\mathbb{R}^n}{\text{maximize}} \quad \inf_{\lambda\geq 0} L(x,\lambda). \tag{3.5}$$

**SAC-Lagrangian** SAC-Lagrangian (SAC-Lag) (Ha et al., 2020) applies the Lagrange multiplier framework to solve constrained Markov Decision Processes (CMDPs) within the Soft Actor-Critic (SAC) paradigm (Haarnoja et al., 2018). The goal is to find a policy $\pi$ that maximizes the entropy-regularized return while ensuring the expected cumulative cost does not exceed a predefined limit $d$. This forms the primal optimization problem for the policy:

$$\begin{aligned}
&\underset{\pi\in\Pi}{\text{maximize}} && \underbrace{\mathop{\mathbb{E}}_{(s_t,a_t)\sim\pi}\sum_{t=0}^{T}\left[\gamma^t R(s_t,a_t) + \alpha H(\pi(s_t))\right]}_{f(\pi)}\\
&\text{subject to} && \underbrace{\mathop{\mathbb{E}}_{(s_t,a_t)\sim\pi}\sum_{t=0}^{T}\left[\gamma^t C(a_t,s_t)\right] - d \leq 0,}_{g(\pi)}
\end{aligned} \tag{3.6}$$

where $H\colon \mathcal{P}(\mathcal{A}) \to \mathbb{R}$ is the entropy function, and $C\colon \mathcal{S} \times \mathcal{A} \to \mathbb{R}$ is the cost function.

Following the dual formulation in (3.5), we can construct the Lagrangian and solve the saddle-point problem

$$\underset{\pi \in \Pi}{\text{maximize}} \quad \inf_{\lambda \geq 0} L(\pi, \lambda).$$

Since this problem has no analytical solution, SAC-Lag employs an actor-critic architecture and solves it using dual gradient descent. The architecture consists of an actor network $\pi(s; \phi)$, a reward Q-network $Q_r(s, a; \theta_r)$, and a cost Q-network $Q_c(s, a; \theta_c)$, where the parameters $\theta_r$ and $\theta_c$ are updated by minimizing the respective TD-errors. The policy parameters $\phi$ and the Lagrange multiplier $\lambda$ are updated iteratively using samples from a replay buffer $\mathcal{D}$. The multiplier $\lambda$ is adjusted via gradient descent on the dual objective

$$J_{\mathrm{d}}(\lambda) = \mathbb{E}_{(s_t, a_t) \sim \mathcal{D}} \left[ \lambda(d - Q_c(s_t, a_t)) \right].$$

This update increases $\lambda$ when the expected cost $Q_c$ exceeds the limit $d$, thereby increasing the penalty for constraint violations in the policy's objective. The actor $\pi_\phi$ is then updated to maximize the Lagrangian, which corresponds to minimizing the actor loss:

$$J(\phi) = \mathbb{E}_{s_t \sim \mathcal{D}} \left[ \alpha \log \pi(s_t; \phi) - Q_r(s_t, \pi(s_t; \phi)) + \lambda Q_c(s_t, \pi(s_t; \phi)) \right]. \tag{3.7}$$

## 4 Approach

### 4.1 Log Barrier Method

The log barrier method provides an alternative to Lagrange multipliers for handling inequality constraints like those in (3.4). It works by augmenting the objective with a barrier function that is only defined on the feasible set, i.e., goes to infinity when approaching the feasible set boundary from the interior, given by

$$\psi(x) = -\frac{1}{\mu} \log(-x), \tag{4.1}$$

where $\mu > 0$ is a parameter that controls the steepness of the barrier. The constrained problem (3.4) is then approximated by the unconstrained problem

$$\underset{x \in \mathbb{R}^n}{\text{maximize}} \quad f(x) - \psi(g(x)) = f(x) + \frac{1}{\mu} \log(-g(x)).$$

The barrier term is negligible when the solution is far from the constraint boundary ($g(x) \ll 0$) but approaches $-\infty$ as the boundary is neared ($g(x) \to 0^-$), effectively preventing the optimizer from leaving the feasible set. The key advantage is that it replaces the non-differentiable hard constraint with a smooth, differentiable penalty.

### 4.2 Linear Smoothed Log Barrier Function

A critical limitation of the standard log barrier (4.1) is that it is undefined for points not in the interior of the feasible set, e.g. consider the problem (3.4), the barrier function $\psi(g(x))$ is undefined for $x \in \mathbb{R}^n$ with $g(x) \geq 0$. This poses a significant challenge in deep reinforcement learning, where: 1. Gradient-based optimizers may temporarily step into the infeasible region during an update. 2. The policy network is often randomly initialized, resulting in an unsafe policy at the start of training.

Directly clipping the cost (define $g(x) = 0$ if $g(x) > 0$) would cause gradient saturation and prevent the agent from learning to escape unsafe regions. To address this, we adopt the linear smoothed log barrier function proposed by Kervadec et al. (2019), defined as:

$$\tilde{\psi}(x) = \begin{cases} -\frac{1}{\mu} \log(-x) & \text{if } x \leq -\frac{1}{\mu^2} \\ \mu x - \frac{1}{\mu} \log(\frac{1}{\mu^2}) + \frac{1}{\mu} & \text{otherwise.} \end{cases} \tag{4.2}$$

As shown in Fig. 1, the function (4.2) is continuous and differentiable everywhere. It preserves the logarithmic barrier within the feasible region but transitions to a linear penalty for constraint violations. This allows meaningful gradients to flow even from unsafe states, guiding the policy back towards feasibility without numerical instability.

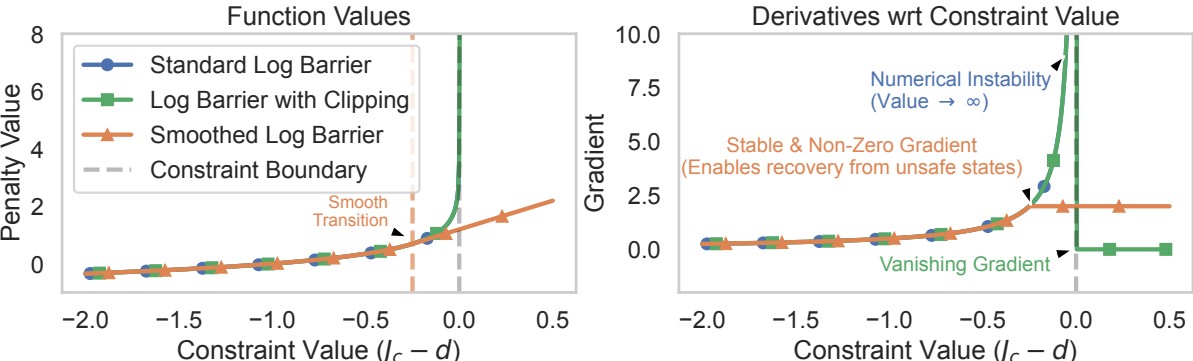

Figure 1: Comparison of different barrier functions and their derivatives. The left figure shows the penalty value, while the corresponding gradient is shown in the right figure. The standard log barrier is numerically unstable for non-negative constraint values ($x \geq 0$). Clipping the input (or gradient) solves this but results in a vanishing gradient for unsafe states ($x > 0$), halting learning. Our proposed smoothed barrier remains well-defined and provides a stable, non-zero gradient, enabling the policy to recover from constraint violations. The barrier parameter is set to $\mu = 2.0$ for visualization.

### 4.3 CSAC-LB

Our method, CSAC-LB, integrates the linear smoothed log barrier (4.2) into the SAC framework to solve the constrained RL problem (3.6). Although the smoothed log barrier function provides a mechanism for learning a safe policy, its effectiveness relies on the cost critic accurately estimating the policy's expected constraint violations. If the cost critic underestimates the policy's cost, recovery from an unsafe policy can be slow and unreliable. van Hasselt et al. (2016) identified an overestimation bias in Q-learning and proposed taking the minimum across multiple Q-networks to mitigate it. A similar issue arises in the constrained setting; during actor optimization, there is a bias that can lead to an underestimation of constraint violations. This is critical for the policy update, as it may hinder the agent's ability to recover from unsafe policies.

To address this problem, our architecture employs twin Q-networks for both the reward and cost critics. For the reward critic, we follow the standard practice from Double DQN of taking the minimum of the Q-values to reduce overestimation. For the cost critic, however, underestimation is the primary concern, as it could lead the agent to select unsafe actions. Therefore, to ensure a conservative and pessimistic estimate of the expected cumulative cost, we take the **maximum** of the two cost Q-values. While other methods (Ji et al., 2024) may use a single cost critic, our double-critic approach for costs is designed to explicitly mitigate the underestimation of costs.

A further challenge in RL is that the cost critic can become overly conservative due to biases in replay data and the approximation errors. To add flexibility and prevent the non-zero gradients of the barrier function from causing suboptimal reward-seeking, we introduce a tunable offset $\delta \geq 0$ (where $\delta = 1$ is used for all the experiments in this paper). Our final barrier function, $\tilde{\psi}$, is applied to the output of the cost critic when training the actor, i.e.,

$$\tilde{\psi}(Q_c(s,a)) = \tilde{\psi}(\max_{i=1,2} Q_{c,i}(s,a) - d - \delta). \tag{4.3}$$

The combination of the fast-acting log barrier penalty and the pessimistic cost estimation provides a strong, immediate corrective signal when approaching or exceeding constraint boundaries. This allows CSAC-LB to recover from unsafe states more rapidly and reliably than methods relying on slower-adapting Lagrange multipliers or optimistic cost estimates. The complete procedure is detailed in Algorithm. 1.

The critic networks $Q_r$ and $Q_c$ are trained by minimizing their respective mean-squared TD errors, where the actor loss is given by replacing the last term in (3.7) for SAC-Lagrangian, i.e.,

$$J(\phi) = \mathbb{E}_{s_t \sim \mathcal{D}}\left[\alpha \log \pi(s_t; \phi) - Q_r(s_t, \pi(s_t; \phi)) + \tilde{\psi}(Q_c(s_t, \pi(s_t; \phi)))\right]. \tag{4.4}$$

## 4.4 Theoretical Analysis: Safety and Performance Guarantees

Our theoretical analysis provides two key guarantees for the CSAC-LB algorithm: a guarantee on the satisfaction of the safety constraints and a bound on the suboptimality of the final policy's performance. A detailed proof can be found in the Appendix. A.4.

**Theorem 1** (Suboptimality Guarantee). *Let $p^*$ be the optimal point of the reinforcement learning problem under CMDP given by (3.3), i.e.,*

$$p^* = \inf_{\pi \in \Pi}\{-J_R(\pi) \mid J_{C_i}(\pi) - d_i \leq 0, \ i = 1, \ldots, m\}, \tag{4.5}$$

*where $J_R(\pi)$ is the expected return given by (3.1) and $J_{C_i}(\pi)$ is the expected cumulative cost for the i-th constraint, given by (3.2). We also assume that the reward and cost functions $R(s,a)$, $C(s,a)$ are bounded. Let $\pi^*$ be the optimal policy found by CSAC-LB, i.e.,*

$$\pi^* = \operatorname*{argmin}_{\pi \in \Pi}\left(-J_R(\pi) + \sum_{i=1}^{m} \tilde{\psi}(J_{C_i}(\pi) - d_i - \delta_i)\right), \tag{4.6}$$

*where the function $\tilde{\psi}$ is the linear smoothed log barrier (4.2), and $\delta_i \geq 0$ is the offset for the i-th constraint. If the constrained problem (3.3) is strictly feasible, (i.e., $\exists \pi \in \Pi$ such that $J_{C_i}(\pi) < d_i$ for all $i = 1, \ldots, m$) and the functions $-J_R(\pi)$, $J_{C_i}(\pi)$ are convex, then the suboptimality bound of the solution $\pi^*$ is given by*

$$-J_R(\pi^*) - p^* \leq \frac{m}{\mu},$$

*where $m$ is the number of constraints and $\mu > 0$ is the barrier parameter.*

**Theorem 2** (Bounded Constraint Violation). *Let $\pi^*$ be the optimal policy found by CSAC-LB, given by (4.6), with some hyperparameter $\mu$ for the barrier function (4.2). Suppose the reward and cost functions are bounded and the original constrained reinforcement learning problem (3.3) is strictly feasible, we then have*

$$\lim_{\mu \to \infty} J_{C_i}(\pi^*) \leq d_i + \delta_i$$

*for all $i = 1, \ldots, m$, i.e., the constraint violation $J_{C_i}(\pi^*) - d_i$ is at most $\delta_i$ as $\mu \to \infty$.*

Theorem 1 shows that CSAC-LB is a formal approximation of the true constrained problem, with a suboptimality gap of $m/\mu$ that can be controlled. This reveals the fundamental trade-off of the barrier parameter $\mu$: a larger $\mu$ reduces the theoretical performance gap by creating a steeper penalty landscape, but such sharp penalties can destabilize the neural network's optimization process as shown in Fig. 7. When the barrier parameter $\mu$ is set to be too large, the agent may struggle with exploring the boundary area and consequently learn a conservative policy. We provide an empirical analysis in our ablation study in Sec. A.2. It is important to note that the convexity assumption required by our formal guarantees is generally not met in deep reinforcement learning. However, as our empirical results demonstrate (see Fig. 4), CSAC-LB still learns robustly safe and high-performing policies in practice. Theorem 2 explains the role of the offset $\delta$ when $\mu \to \infty$. In practice, while adjusting $\delta$ is functionally similar to changing the cost limit $d$, its true utility comes from its interaction with the shape of our smoothed barrier function. By shifting the barrier's steep logarithmic region from $d$ to $d+\delta$, it creates a "soft" linear penalty zone for minor constraint violations. This is particularly beneficial when the cost limit is very strict (e.g., $d = 0$), as it prevents the immediate, harsh penalty at the boundary and thus smooths the training process.

## 5 Experiments

### 5.1 Experiment Setup

**Environment**  To evaluate the generalization of our algorithm CSAC-LB and the other baselines, we conduct experiments in 10 tasks introduced by Luo & Ma (2021) and Ji et al. (2023). These high-risk, high-reward tasks cover a wide spectrum ranging from 2D navigation task to continuous control tasks and from simple pendulum Tilt task to high-dimensional humanoid locomotion task with speed limits. All tasks are depicted in the Fig. 2. Notably, there are four tasks derived from Pendulum and Inverted Pendulum. A detailed introduction can be found in the Appendix. A.1.

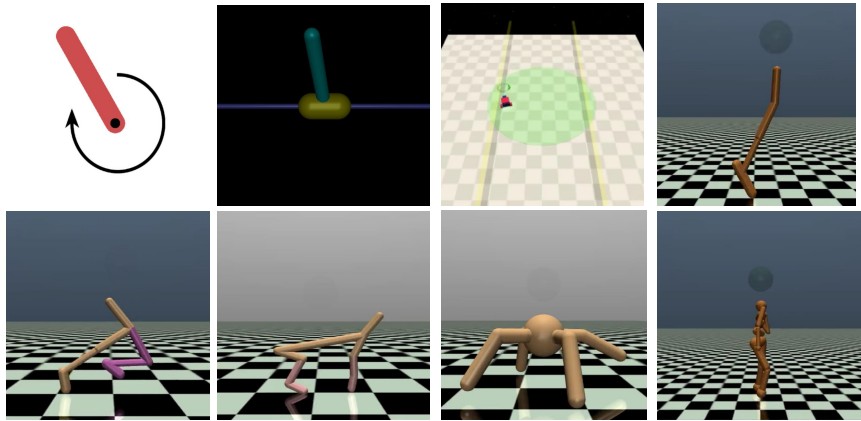

Figure 2: **Visualization of the experimental environments.** The environments include Pendulum and Inverted Pendulum, with two distinct tasks for each as defined in Luo & Ma (2021)

**Baseline**  We evaluate CSAC-LB against fix prominent baseline methods, chosen to represent different approaches to constrained reinforcement learning: 1) **SAC** with Reward Shaping, a standard SAC implementation (Haarnoja et al., 2018) where a penalty term is subtracted from the environment reward upon constraint violation (i.e. the modified reward function $\tilde{R}(s,a) = R(s,a) - \text{penalty factor} \cdot C(s,a)$, where R(s,a) is the true reward function from the environments). The setup follows the methodology from Luo & Ma (2021). 2) **SAC-Lagrangian (SAC-Lag)**, an off-policy algorithm that extends SAC using the Lagrangian method to handle constraints (Achiam, 2021), as detailed in Section 3.2. 3) **Worst-Case SAC (WCSAC)**, an extension of SAC-Lagrangian that incorporates the Conditional Value-at-Risk (CVaR) of the cost distribution as its risk measure (Yang et al., 2021). 4) **CPO**, a widely-cited on-policy algorithm that enforces constraints using trust region optimization, providing strong theoretical guarantees (Achiam et al., 2017). 5) **APPO**, a variation of PPO with Lagrangian method by augmenting the policy loss with a quadratic deviation term (Dai et al., 2023). Recognizing that the performance of SAC with reward shaping and WCSAC is highly sensitive to their respective penalty factors and risk levels, we conducted a hyperparameter search to ensure a fair and robust comparison. Based on this search (summarized in Fig. 4), we selected optimal values of 6 for the SAC penalty factor and 0.5 for the WCSAC risk level. For clarity in our results, we refer to these tuned baselines as SAC-6 and WCSAC-0.5, respectively. All algorithms were re-implemented and trained with the same hyperparameters, as detailed in Table. 2.

### 5.2 Experiment Results

**Comparison with Baselines**  Fig. 3 depicts the learning curves of the performance and constraint satisfaction of all algorithms across environments. We set a challenging cost limit of $d = 0$ for the simpler tasks (shown in the upper row) to test numerical stability, and $d = 25$ for the more complex locomotion tasks (in the lower row). The results show that CSAC-LB consistently achieves high returns while adhering to the specified safety constraints across all environments. For example, in the Walker2d and Ant tasks, CSAC-

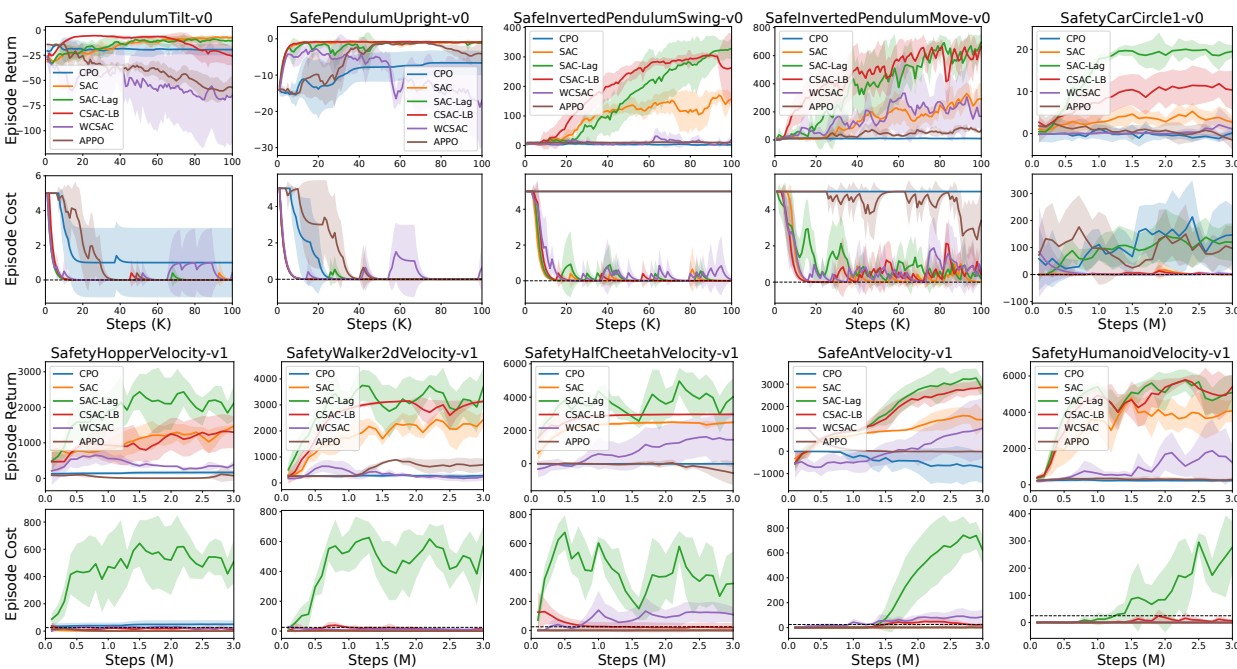

Figure 3: **Training performance and safety over time.** The top and bottom panels show the episodic return and cost of constraint violations, respectively, versus training steps. Solid lines represent the mean evaluation performance across 5 independent random seeds, while the shaded regions denote one standard deviation. The dashed horizontal line in the cost plot indicates the cost limit for the task.

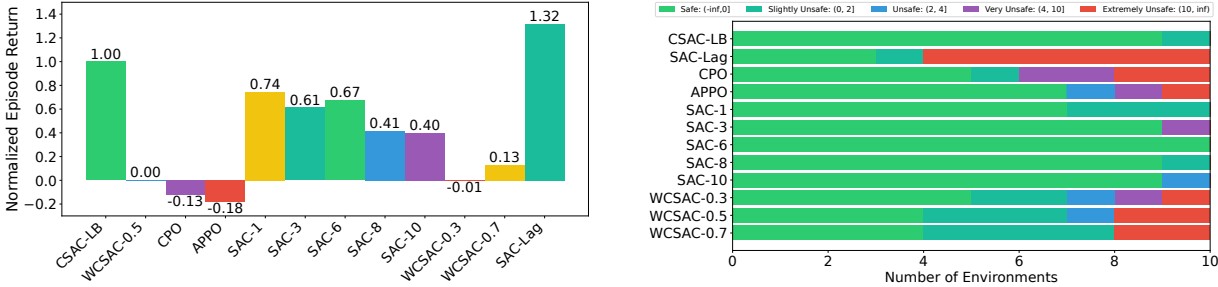

Figure 4: **Aggregate performance and safety analysis across all environments**. **(Left)** Normalized average return. The performance of each algorithm is scaled such that our method (CSAC-LB) scores 1 and the conservative baseline (WCSAC-0.5) scores 0, highlighting relative improvement. **(Right)** Distribution of constraint satisfaction. Evaluation runs are categorized into discrete safety levels based on their final cumulative cost. The bars show the number of environments for each algorithm that fall into each category. Both plots show results averaged over the final 5 evaluation episodes across 5 independent random seeds.

LB achieves approximately 24% and 150% higher final returns, respectively, than the next best-performing safe baseline, SAC. In contrast, other methods demonstrate clear limitations. SAC-Lag often achieves high returns but at the cost of significant constraint violations, with average costs reaching up to 700 in the locomotion tasks. WCSAC and SAC generally remain safe but yield substantially lower returns. Moreover, WCSAC fails to learn safe policies in the Ant and HalfCheetah environments. CPO consistently underperformed, exhibiting extremely conservative policy in locomotion tasks. However, we do note a performance degradation for CSAC-LB in the final stages of the PendulumTilt task and CSAC-LB is no longer among the best-performing safe policies. We hypothesize this occurs because once the policy converges, the entropy

bonus from SAC can encourage unnecessary exploration near the boundary, leading to minor violations and a subsequent over-correction by the barrier function. As we discuss in Appendix A.3, this can be mitigated by adjusting the offset parameter $\delta$ to prevent the convergence. We further conduct a statistics of the overall performance of different algorithms as shown in Fig. 4. The left panel presents the normalized average return, which is scaled to highlight the performance improvement over a conservative but suboptimal baseline (WCSAC-0.5). The results show that despite scoring at 1.32, SAC-Lag severely violates constraints. Whereas, other methods, such as the SAC variants, achieve significantly lower relative scores of 0.75 at their highest, 25% less than our methods. All APPO, WCSAC and CPO runs performed poorly. The right panel provides the corresponding safety analysis, showing the distribution of constraint satisfaction across the ten environments. CSAC-LB demonstrates a high degree of reliability, successfully solving 9 out of 10 tasks with safe policies. In only one task (Inverted PendulumMove) did it produce a slightly unsafe policy, with a minor constraint violation of 0.6. In contrast, the high performance of SAC-Lag is shown to be at a cost, as it results in unsafe policies in 7 out of the 10 environments. While SAC variants are predominantly safe, their low scores in the left panel confirm that this safety comes at a significant performance cost.

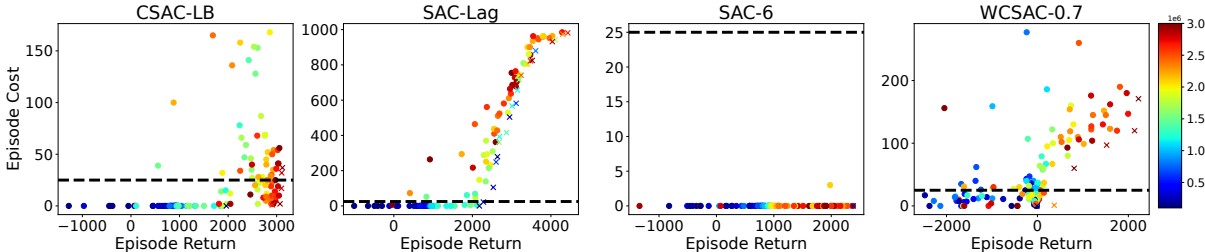

Figure 5: **Evolution of learned policies for different algorithms on the SafeAntVelocity task.** Each point represents the return and cost of a single evaluation episode, with color indicating training progress (from blue/early to red/late). The black dashed line indicates the cost limit $d$, while the crosses mark the Pareto front. The plot illustrates that our method, CSAC-LB, learns to safely explore the regime around the cost limit while efficiently converging towards high-return and safe solutions. The baseline methods showcase clear failure modes: SAC-Lag and WCSAC are unsafe, frequently breaching the cost limit, while SAC-6 learns an overly safe yet conservative policy.

**Analysis of Exploration Behavior**   To provide a qualitative understanding of the advantage of CSAC-LB in terms of exploration, Fig. 5 visualizes the training behaviors of different algorithms on the SafeAntVelocity task. Each point represents the return and cost of a policy during an evaluation episode, with its color indicating the training step at which the evaluation took place. Initially, CSAC-LB adopts a conservative policy, operating well within the safe margin to achieve a return of approximately 1000. As training progresses, it briefly explores beyond the cost limit of $d = 25$, a behavior enabled by the stable linear penalty of our smoothed barrier, which allows the agent to find higher-reward states without destabilizing. In the final training stages, its policies converge to the constraint boundary, with most later-stage policies (red points) clustering near the cost limit. By precisely exploring this boundary, CSAC-LB successfully pushes the Pareto-optimal frontier, achieving a high final return of approximately 3000 while satisfying the safety constraint. The baseline methods, in contrast, exhibit less effective strategies. SAC-Lag and WCSAC display a high variance in their exploration; their evaluation points are scattered across the plot and different colored points all mix up together, indicating a failure to converge to a stable policy. Critically, numerous later-stage policies (red points) significantly exceed the safety limit, demonstrating unreliable constraint satisfaction. Conversely, SAC-6 represents the opposite extreme: it is overly conservative. While it quickly learns a safe policy, it fails to explore the space near the boundary, causing it to prematurely converge to a suboptimal return of around 2000. Overall, CSAC-LB achieves a good balance between safety and performance. This success is attributed to our smoothed log barrier design, which permits the agent to effectively explore the crucial region near the constraint boundary. Compared to the baseline algorithms, CSAC-LB allows for gathering more information from the crucial region near the constraint boundary.

**Approximation Errors of Cost Critic**    To validate our design choice of using a twin cost critic network, we performed an ablation study on the number of cost critics ($N$) used in our architecture. The results, presented in Fig. 6, highlight the critical issue of the critic network's value estimation accuracy and its effects. We use SafeCarCircle1-v0 task with its cost limit $d = 0$. This creates extra challenges for the cost critics. The single critic network ($N = 1$) fails to maintain safety, consistently showing high episodic costs. This can be attributed to the value underestimation. The left panel shows the single critic predicting large negative Q-values, a clear approximation error given that episodic costs are always non-negative. This leads the policy to incorrectly evaluate dangerous states as safe, resulting in the observed constraint violations. The introduction of a second critic ($N = 2$) and taking the maximum of their Q-values rectifies this issue. This approach introduces a pessimistic bias that effectively counteracts the underestimation tendency, stabilizing the cost estimate at a value greater than 0, mostly, and leading to safe behaviour in the end. While using more critics ($N = 3, 4$) also ensures safety, these configurations become overly pessimistic and prevent the agent from discovering higher-reward policies, causing their final returns to be significantly lower than the $N = 2$ case. Therefore, we conclude that using two cost critics provides the best balance: it introduces sufficient pessimism to ensure stable and safe learning while avoiding the overly conservative behaviour with more critics.

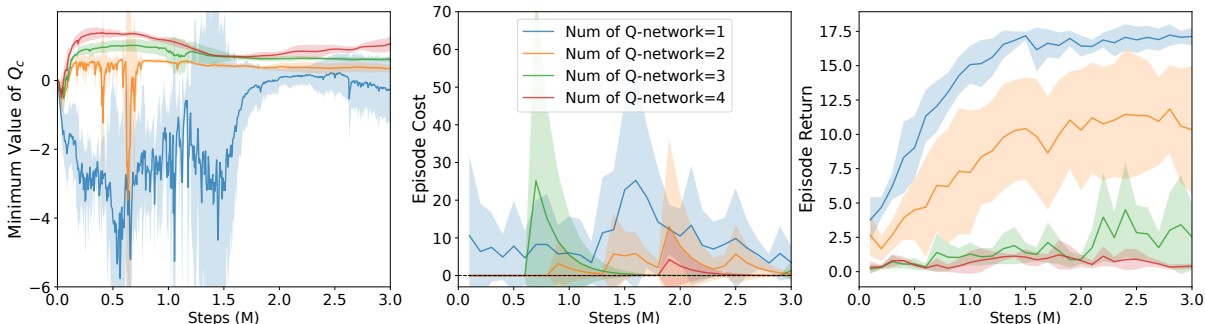

Figure 6: **Training curves of different number of cost critic networks $Q_c$ in SafeCarCircle1-v0 task.** The left panel shows the minimum Q-value predicted by the target cost critic(s) during batch updates, while the middle and right panels show the corresponding episodic returns and costs. Notably, the single critic configuration demonstrates significant underestimation. Despite only non-negative costs occurring in the environment, it predicts negative Q-values. This flawed estimation leads the agent to perceive unsafe actions as safe, resulting in poor constraint satisfaction.

## 6    Limitations

While CSAC-LB demonstrates robust performance, it is important to acknowledge its limitations. The scope of our evaluation was conducted entirely in simulation; deploying CSAC-LB in real-world systems would require addressing additional challenges, such as the sim-to-real gap. Our formal guarantees on suboptimality and constraint satisfaction rely on idealized assumptions, including convexity, which do not hold in the general deep RL setting. Thus, the theory serves as valuable guidance for the algorithm's design, while its practical effectiveness is confirmed by our empirical results.

## 7    Conclusion

In this work, we introduced Constrained Soft Actor-Critic with Log Barrier (CSAC-LB), a novel off-policy algorithm designed to overcome the critical challenges of training instability and sample inefficiency in safe reinforcement learning. Our approach successfully integrates a linear smoothed log barrier function into the actor's objective. This core innovation provides a numerically stable gradient signal that allows the agent to learn effectively from constraint violations, enabling rapid and reliable recovery from unsafe states without the oscillations seen in Lagrangian methods or the gradient issues of traditional barriers. To further

improve its safety, CSAC-LB employs a pessimistic double-critic architecture for the cost function, which provides a conservative estimate of expected costs and robustly prevents the policy from exploiting critic underestimation errors. These design choices are supported by a theoretical analysis that provides guarantees on the policy's suboptimality and constraint violation.

Our extensive empirical evaluation across ten diverse and challenging benchmark tasks demonstrated the clear superiority of CSAC-LB. The algorithm consistently achieved a superior balance of high returns and strict constraint satisfaction, succeeding where other prominent methods failed, all while avoiding the need for pre-training, reward shaping, or a learned dynamics model. While baselines either violated constraints in pursuit of high rewards or settled for overly conservative, suboptimal policies, CSAC-LB learned to proficiently and safely explore the crucial boundary of the feasible region. This allowed it to discover policies that are both high-performing and safe, highlighting the effectiveness of its design.

The robust performance and stable learning dynamics of CSAC-LB establish it as a powerful and practical solution for CMDPs. Future work could proceed along several avenues. One direction is to enhance the core algorithm by exploring adaptive mechanisms for the barrier's parameters $(\mu, \delta)$ or by integrating more advanced critic representations to tackle complex, high-dimensional problems. A second, broader direction is to extend our approach to more complex problem domains, such as real-world applications, multi-agent systems, environments with time-varying constraints, or problems involving stochastic safety specifications. Overall, CSAC-LB provides a reliable framework for developing intelligent agents that can operate effectively and safely in safety-critical domains.

## 8 Acknowledgement

This work is funded by the Deutsche Forschungsgemeinschaft (DFG, German Research Foundation) – Project-ID 499552394 – SFB 1597 and the German Federal Ministry for the Environment, Nature Conservation and Nuclear Safety (BMU) on the basis of a resolution of the German Bundestag as part of the 'KI-Leuchtturm' project 'Intelligence for Cities' (I4C). This project is also funded by the German Research Foundation (DFG) under grants 417962828, 428605208 and 539134284.

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

# A    Appendix

## A.1    Environment Descriptions

This evaluation uses a diverse set of reinforcement learning environments designed to test different aspects of algorithm performance and safety.

### Low-dimensional Tasks

These four tasks are based on classic control problems (Luo & Ma, 2021), but are modified to create high-risk, high-reward scenarios where the agent must operate near a safety boundary to succeed.

**Upright**   Based on Pendulum, the agent must keep the pole perfectly upright. The reward is maximized by minimizing the tilt angle, while a safety constraint prevents the pole from falling.

**Tilt**   Also based on Pendulum, this task requires the agent to balance the pole at a large, specific tilt angle, making exploration near the safety boundary necessary for high rewards.

**Move**   Based on CartPole, the agent is rewarded for moving the cart far from its starting point while keeping the pole nearly vertical and within tight safety bounds.

**Swing**   Also based on CartPole, the agent is rewarded for swinging the pole to a large angle, requiring it to manage its position while avoiding falling.

### Safety-Gymnasium Benchmark

These tasks are from the standard Safety-Gymnasium benchmark (Ji et al., 2023), testing navigation and locomotion under explicit safety constraints.

**CarCircle-v1**   A car-like robot must navigate a circular path. The reward encourages the agent to drive as fast as possible and be as close as possible to the circle, while a cost is incurred for crossing the track boundaries.

**Velocity-Constrained Locomotion Tasks**   This suite of environments (Hopper, Walker2D, HalfCheetah, Ant, Humanoid) features common MuJoCo agents with a shared objective. In each task, the agent is rewarded for maximizing its forward velocity while being constrained by a safety limit set to 50% of the velocity achieved by vanilla PPO in the original task.

## A.2    Ablation Study

To evaluate the sensitivity of CSAC-LB to its key hyperparameters, we conduct an ablation study on the barrier parameter $\mu$ and the offset $\delta$. The results, presented in Fig. 7, show the training performance across different hyperparameter settings in two representative environments: the numerically challenging SafeCarCircle1-v0 and the complex locomotion task SafetyHopperVelocity-v1.

In CarCircle task, CSAC-LB demonstrates high robustness. While larger values of $\mu$ (e.g., 10) introduce minor training fluctuations, the final performance remains consistent across all tested values. This suggests that for tasks with simpler dynamics, the exact steepness of the barrier is less critical. The offset $\delta$ shows a more significant effect. Although most values yield similar outcomes, setting $\delta = 3.0$ leads to a notable improvement in the final return (from approximately 10 with our default settings to 17) while maintaining zero cost. This highlights the utility of $\delta$ in creating a "soft" linear penalty zone around the constraint boundary, preventing the policy from becoming overly conservative, especially when the cost limit is strict ($d = 0$ in this case).

Conversely, the HopperVelocity task is more sensitive to the choice of $\mu$. As shown, lower values ($\mu = 3, 4$) achieve a high return of approximately 1500. However, increasing $\mu$ to 6 or 10 causes a significant performance

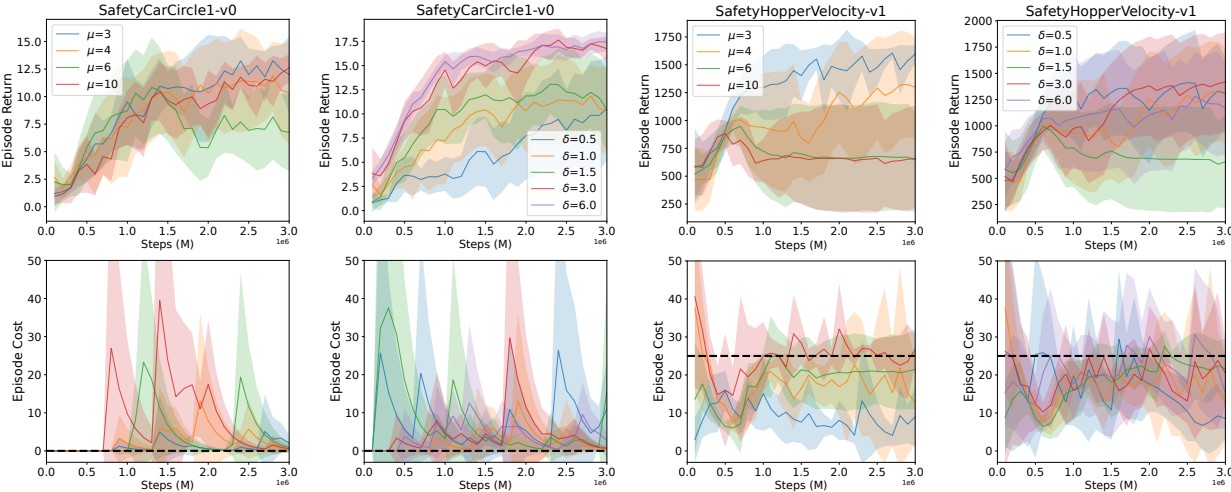

Figure 7: **Ablation study on the barrier parameter $\mu$ and offset $\delta$.** Training curves for episodic return and cost are shown for SafeCarCircle1-v0 and SafetyHopperVelocity-v1. The dashed horizontal line in the cost plots indicates the cost limit $d$. The results highlight the fundamental trade-off of the barrier parameter $\mu$: while the algorithm is broadly robust, excessively large values can destabilize training in complex tasks like Hopper. The offset $\delta$ provides a mechanism for fine-tuning, improving performance in the strict CarCircle task without compromising safety.

drop to around 600. This aligns with the trade-off identified in our theoretical analysis (Theorem 1): a large $\mu$ creates a steep penalty landscape that, while theoretically optimal, can destabilize the neural network's optimization process in complex control tasks, leading to a suboptimal policy. Notably, all tested $\mu$ values successfully keep the cost below or very near the limit ($d = 25$). The algorithm is largely robust to the choice of $\delta$ in this environment, with all settings converging to a similar high-performance, safe policy.

Overall, this study demonstrates the general robustness of CSAC-LB. It suggests that $\mu = 4$ provides a good balance between theoretical optimality and practical stability, while $\delta$ can be used as an effective parameter for fine-tuning performance, particularly in tasks with very strict constraints.

## A.3 Failure Case Analysis

In our main results (Fig. 3), we identified a late-stage performance degradation for CSAC-LB in the PendulumTilt task. Figure 8 provides a deeper analysis of this behavior and demonstrates how the offset parameter $\delta$ serves as an effective remedy.

The issue stems from a subtle interaction between SAC's automatic entropy tuning and the sharp constraint boundary enforced by our log barrier. With the default $\delta = 1$, the policy quickly converges to a near-optimal, low-entropy state. As shown in the left panel of Fig. 8, this causes the policy's entropy to drop below SAC's fixed target entropy. Consequently, the automatic tuning mechanism dramatically increases the temperature parameter $\alpha$ to encourage more exploration. This forced, stochastic behavior near a strict constraint boundary inevitably leads to constraint violations (right panel), which are then heavily penalized by the barrier function. This penalty, in turn, causes the policy to become overly conservative, leading to the observed collapse in performance (middle panel).

This analysis provides a clear guide for practitioners on managing this behavior. The most direct solution is to increase the offset parameter $\delta$ (e.g., to 2 or 3), which creates a wider "buffer zone" around the true constraint limit $d$. This allows the policy to maintain a slightly higher entropy without triggering the harshest region of the logarithmic penalty, causing the temperature $\alpha$ to remain stable and averting the performance collapse. An alternative, more advanced approach is to manually lower the target entropy for SAC, which directly reduces the pressure for forced exploration that causes the issue. While these tuning strategies are

effective, exploring fully automated methods for managing this trade-off is a promising direction for future work.

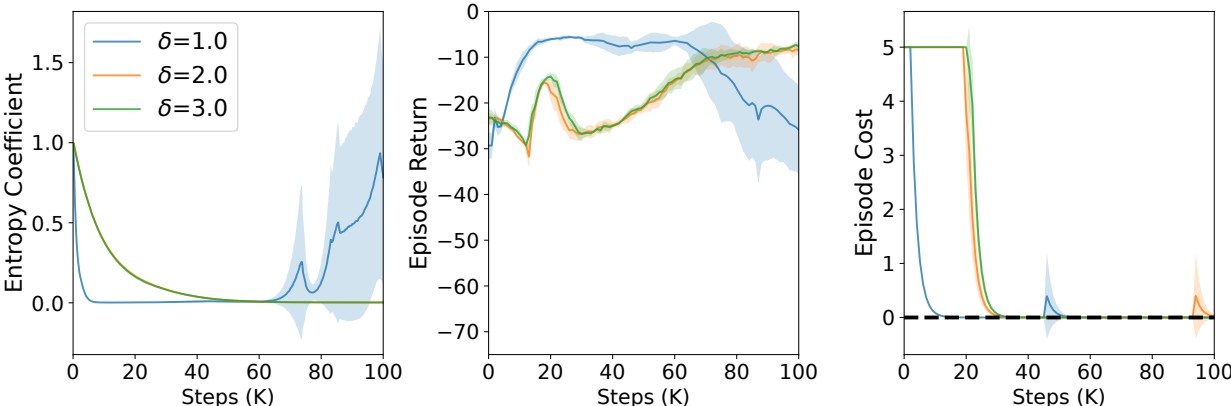

Figure 8: **Analysis of the performance degradation in SafePendulumTilt-v0 and its mitigation using the offset $\delta$.** The figure displays the evolution of the SAC entropy coefficient $\alpha$ (left), episodic return (middle), and cost (right) for different values of the offset parameter $\delta$. With the default $\delta = 1$, the policy converges successfully but then degrades as the entropy coefficient rises to enforce exploration, leading to increased costs. Increasing $\delta$ to 2 or 3 stabilizes the training by creating a larger buffer around the constraint boundary, preventing this late-stage collapse and maintaining a high-performing, safe policy.

### A.4    Proof of Theorem 1

*Proof.* The Lagrangian for the constrained problem (3.3) is given by

$$L(\pi, \lambda) = -J_R(\pi) + \sum_{i=1}^{m} \lambda_i (J_{C_i}(\pi) - d_i), \tag{A.1}$$

where $\lambda = (\lambda_1, \dots, \lambda_m) \in \mathbb{R}_+^m$ is the Lagrange multiplier. The corresponding dual function is then

$$g(\lambda) = \inf_{\pi \in \Pi} L(\pi, \lambda). \tag{A.2}$$

Since the policy $\pi^*$ given by (4.6) is the optimal policy found by CSAC-LB via solving an unconstrained problem, the first-order optimality condition requires the gradient of the objective to be zero, i.e.,

$$-\nabla J_R(\pi^*) + \sum_{i=1}^{m} \tilde{\psi}'(J_{C_i}(\pi^*) - d_i - \delta_i) \nabla J_{C_i}(\pi^*) = 0. \tag{A.3}$$

We can then construct a Lagrange multiplier $\lambda^*$ that is dual feasible by setting each of its component as:

$$\lambda_i^* = \tilde{\psi}'(J_{C_i}(\pi^*) - d_i - \delta_i) \quad i = 1, \dots, m. \tag{A.4}$$

(We have $\lambda^* \succeq 0$ since the derivative $\tilde{\psi}'$ is always nonnegative.) Substituting (A.4) into (A.1), and then taking the first derivative with respect to $\pi$ for $L(\pi, \lambda^*)$, we have

$$\nabla_\pi L(\pi, \lambda^*) = -\nabla J_R(\pi) + \sum_{i=1}^{m} \tilde{\psi}'(J_{C_i}(\pi^*) - d_i - \delta_i) \nabla J_{C_i}(\pi).$$

By the optimality condition (A.3), it follows immediately that $\nabla_\pi L(\pi^*, \lambda^*) = 0$. Since the Lagrangian is convex by our assumption, this implies that $\pi^*$ minimizes the Lagrangian $L(\pi, \lambda^*)$ for this specific choice of

$\lambda^*$ given by (A.4). We can therefore evaluate the dual function (A.2) at $\lambda^*$:

$$g(\lambda^*) = \inf_{\pi} L(\pi, \lambda^*) = L(\pi^*, \lambda^*)$$

$$= -J_R(\pi^*) + \sum_{i=1}^{m} \lambda_i^* (J_{C_i}(\pi^*) - d_i)$$

$$= -J_R(\pi^*) + \sum_{i=1}^{m} \tilde{\psi}'(J_{C_i}(\pi^*) - d_i - \delta_i)(J_{C_i}(\pi^*) - d_i). \tag{A.5}$$

Notice that by weak duality, the primal optimal value $p^*$ is bounded by the dual function, i.e., $p^* \geq g(\lambda^*)$. Together with (A.5), we have the following suboptimality gap:

$$-J_R(\pi^*) - p^* \leq -\sum_{i=1}^{m} \tilde{\psi}'(J_{C_i}(\pi^*) - d_i - \delta_i)(J_{C_i}(\pi^*) - d_i). \tag{A.6}$$

Now we need to find a uniform upper bound for the summation on the right-hand side of (A.6). Let $x_i = J_{C_i}(\pi^*) - d_i$, $i = 1, \ldots, m$. We then need to bound the term

$$h(x_i) = -\tilde{\psi}'(x_i - \delta_i)x_i.$$

- **Case 1: Logarithmic Region.** Suppose $x_i - \delta_i \leq -1/\mu^2$, by (4.2), we have

$$\tilde{\psi}'(x_i - \delta_i) = -1/(\mu(x_i - \delta_i)).$$

  Then $h(x_i)$ can be expressed as:

$$h(x_i) = -x_i \left( \frac{-1}{\mu(x_i - \delta_i)} \right) = \frac{x_i}{\mu(x_i - \delta_i)} = \frac{1}{\mu} \left( 1 + \frac{\delta_i}{x_i - \delta_i} \right).$$

  Since $\delta_i \geq 0$ and the denominator $x_i - \delta_i$ is negative and non-zero, we have $\frac{\delta_i}{x_i - \delta_i} \leq 0$. Hence, we conclude that

$$h(x_i) \leq \frac{1}{\mu}.$$

- **Case 2: Linear Region.** Suppose $x_i - \delta_i > -1/\mu^2$, by (4.2), we have

$$\tilde{\psi}'(x_i - \delta_i) = \mu.$$

  Then $h(x_i)$ can be expressed as:

$$h(x_i) = -\mu x_i.$$

  Recall that $x_i > \delta_i - 1/\mu^2$ by our assumption, and hence, we have

$$h(x_i) = -\mu x_i < -\mu \left( \delta_i - \frac{1}{\mu^2} \right) = \frac{1}{\mu} - \delta_i \mu \leq \frac{1}{\mu}.$$

In both cases, each individual term in the summation of (A.6) is bounded above by $1/\mu$. We can therefore bound the entire sum as

$$-\sum_{i=1}^{m} \tilde{\psi}'(J_{C_i}(\pi^*) - d_i - \delta_i)(J_{C_i}(\pi^*) - d_i) \leq \sum_{i=1}^{m} \frac{1}{\mu} = \frac{m}{\mu}. \tag{A.7}$$

Substituting (A.7) back into (A.6), we conclude that

$$-J_R(\pi^*) - p^* \leq \frac{m}{\mu}, \tag{A.8}$$

which ends the proof. □

This result shows that the suboptimality of the solution is proportional to the number of constraints $m$ and inversely proportional to the barrier parameter $\mu$. The gap can therefore be made arbitrarily small by choosing a sufficiently large $\mu$.

### A.5 Proof of Theorem 2

*Proof.* According to the expression of the linear smoothed log barrier function (4.2), we have $\tilde{\psi} \to \psi$ as $\mu \to \infty$ (where $\psi$ is the original log barrier function given by (4.1)). Let $I_- : \mathbb{R} \to \mathbb{R} \cup \{\infty\}$ be the indicator function for the nonpositive reals, given by

$$I_-(x) = \begin{cases} 0 & x \leq 0 \\ \infty & x > 0, \end{cases}$$

we have $\psi \to I_-$ as $\mu \to \infty$ (Nesterov & Nemirovskii, 1994; Boyd & Vandenberghe, 2004). Then it follows that $\tilde{\psi} \to I_-$ as $\mu \to \infty$, and hence, the optimization problem corresponding to (4.6), given by

$$\underset{\pi \in \Pi}{\text{minimize}} \quad -J_R(\pi) + \sum_{i=1}^m \tilde{\psi}(J_{C_i}(\pi) - d_i - \delta_i), \tag{A.9}$$

converges to

$$\begin{aligned} \underset{\pi \in \Pi}{\text{minimize}} \quad & -J_R(\pi) \\ \text{subject to} \quad & J_{C_i}(\pi) \leq d_i + \delta_i, \quad i = 1, \dots, m \end{aligned} \tag{A.10}$$

as $\mu \to \infty$. Recall that $\pi^*$ is defined as the optimal point of the problem (A.9) for some $\mu$, and let $\pi_\infty^*$ be the optimal point of the problem (A.10). The optimal point $\pi_\infty^*$ exists since the problem (A.10) is a relaxation of the original constrained reinforcement learning problem (3.3), whose optimal point exists by our assumption. Then it follows from the discussion above that we have $\pi^* \to \pi_\infty^*$ as $\mu \to \infty$. Since the policy $\pi_\infty^*$ satisfies

$$J_{C_i}(\pi_\infty^*) \leq d_i + \delta_i$$

for all $i = 1, \dots, m$, and $J_{C_i}(\pi^*) \to J_{C_i}(\pi_\infty^*)$ as $\pi^* \to \pi_\infty^*$, we conclude that

$$\lim_{\mu \to \infty} J_{C_i}(\pi^*) \leq d_i + \delta_i$$

for all $i = 1, \dots, m$, which ends the proof. $\qquad \square$

Table 1: Environments Overview

| Environment | Base / Agent | Observations | Actions |
|---|---|---|---|
| Upright | Pendulum | 3 | 1 |
| Tilt | Pendulum | 3 | 1 |
| Move | CartPole | 4 | 2 |
| Swing | CartPole | 4 | 2 |
| CarCircle-v1 | Car | 24 | 2 |
| Hopper-Velocity-v1 | Hopper | 11 | 3 |
| Walker2D-Velocity-v1 | Walker2D | 17 | 6 |
| HalfCheetah-Velocity-v1 | HalfCheetah | 17 | 6 |
| Ant-Velocity-v1 | Ant | 105 | 8 |
| Humanoid-Velocity-v1 | Humanoid | 348 | 17 |

Table 2: Hyperparameter Configuration

| Hyperparameter | Value |
|---|---|
| *Common Parameters* | |
| Batch Size | 256 |
| Network Architecture | [256, 256] |
| Discount Factor ($\gamma$) | 0.99 |
| Random Steps | 100 |
| Learning Rate | $1 \times 10^{-3}$ |
| Actor Update Frequency | 1 |
| Critic Update Frequency | 1 |
| Polyak Update Factor | 0.005 |
| Initial Temperature | 1.0 |
| Normalize Reward | Yes |
| *CSAC-LB Parameters* | |
| Offset | 1.0 |
| Log Barrier Factor | 4.0 |
| *WCSAC Parameter* | |
| Damp Scale | 10 |
| *CPO Parameters* | |
| GAE Lambda ($\lambda$) | 0.95 |
| Line Search Max Iterations | 15 |
| CG Max Steps | 15 |
| Normalize Advantage | Yes |

---
**Algorithm 1** Constrained Soft Actor-Critic with Log Barriers (CSAC-LB)

---
1: **Initialize:** Actor parameters $\pi$.
2: **Initialize:** Twin reward critic parameters $\theta_{r,1}, \theta_{r,2}$.
3: **Initialize:** Twin cost critic parameters $\theta_{c,1}, \theta_{c,2}$.
4: **Initialize:** Target networks with $\theta'_{r,i} \leftarrow \theta_{r,i}$ and $\theta'_{c,i} \leftarrow \theta_{c,i}$ for $i = 1, 2$.
5: **Initialize:** Empty replay buffer $\mathcal{D}$.
6: **Hyperparameters:** Cost limit $d$, offset $\delta$, barrier factor $\mu$, discount $\gamma$, entropy coef. $\alpha$, target update rate $\tau$, learning rates $\lambda_\pi, \lambda_Q$.
7: **for** each episode **do**
8:    **for** each environment step **do**
9:       Select action from policy: $a_t \sim \pi_\pi(\cdot|s_t)$.
10:      Execute $a_t$, observe reward $r_t$, cost $c_t$, and next state $s_{t+1}$.
11:      Store transition $(s_t, a_t, r_t, c_t, s_{t+1})$ in $\mathcal{D}$.
12:      Sample a minibatch of transitions $\mathcal{B} = \{(s, a, r, c, s')\}$ from $\mathcal{D}$.
       {— **Critic Updates** —}
       With target policy actions $a' \sim \pi_\pi(\cdot|s')$:
       Compute the reward target (using clipped double-Q):
13:        $y_r \leftarrow r + \gamma \left( \min_{i=1,2} Q'_{\theta'_{r,i}}(s', a') - \alpha \log \pi_\pi(a'|s') \right)$.
       Compute the cost target (using pessimistic double-Q):
14:        $y_c \leftarrow c + \gamma \left( \max_{i=1,2} Q'_{\theta'_{c,i}}(s', a') \right)$.
       Update reward critics by one step of gradient descent on the MSE loss:
15:        $\mathcal{L}(\theta_{r,i}) = \mathbb{E}_{(s,a) \sim \mathcal{B}} \left[ (Q_{\theta_{r,i}}(s, a) - y_r)^2 \right]$ for $i = 1, 2$.
       Update cost critics by one step of gradient descent on the MSE loss:
16:        $\mathcal{L}(\theta_{c,i}) = \mathbb{E}_{(s,a) \sim \mathcal{B}} \left[ (Q_{\theta_{c,i}}(s, a) - y_c)^2 \right]$ for $i = 1, 2$.
       {— **Actor Update** —}
       For actions from the current policy $a_{\text{policy}} \sim \pi_\pi(\cdot|s)$:
17:        Compute pessimistic cost Q-value: $Q_c^{\max}(s, a_{\text{policy}}) = \max_{i=1,2} Q_{\theta_{c,i}}(s, a_{\text{policy}})$.
18:        Compute the log barrier penalty: $P(s, a_{\text{policy}}) = \tilde{\psi}(Q_c^{\max}(s, a_{\text{policy}}) - d - \delta)$.
       Update actor by one step of gradient descent on the objective:
19:        $J(\pi) = \mathbb{E}_{s \sim \mathcal{B}, a \sim \pi_\pi} \left[ -\min_{i=1,2} Q_{\theta_{r,i}}(s, a) + \alpha \log \pi_\pi(a|s) + P(s, a) \right]$.
       {— **Target Network Updates** —}
       Update all target networks using Polyak averaging:
20:        $\theta'_{r,i} \leftarrow \tau\theta_{r,i} + (1 - \tau)\theta'_{r,i}$ for $i = 1, 2$.
21:        $\theta'_{c,i} \leftarrow \tau\theta_{c,i} + (1 - \tau)\theta'_{c,i}$ for $i = 1, 2$.
22:    **end for**
23: **end for**

---

