# OpenReview forum: "Constrained Reinforcement Learning with Smoothed Log Barrier Function"
_TMLR — Accepted by TMLR_

### Review · Reviewer_zicq · 2025-08-05

**Summary Of Contributions:**

This paper proposes CSAC-LB, a model-free, sample-efficient, and off-policy reinforcement learning algorithm designed for safe RL in real-world systems where strict safety constraints must be satisfied during both training and deployment. The key innovation is integrating a linear smoothed log barrier function into the actor's objective, which enforces constraint satisfaction without requiring pre-training or suffering from instability issues common in traditional constrained RL methods. To further mitigate underestimation of constraint violations, the authors employ a pessimistic double-critic architecture for the cost function. Theoretical analysis provides suboptimality and constraint violation bounds, and comprehensive experiments on ten benchmark tasks validate the method’s robustness, safety, and performance advantages over existing baselines.

**Additional Comments:**

### Strengths

1. **Well-Motivated and Practical Problem**

    The paper targets a critical challenge in safe RL, **balancing safety and training stability**, which has direct implications for real-world deployment in safety-critical domains.

2. **Key Innovation: Linear Smoothed Log Barrier**
    - Overcomes the limitations of standard log barrier methods that are undefined outside the feasible set.
    - Provides **non-vanishing gradients**, enabling meaningful updates even from unsafe states and avoiding instability from abrupt gradient saturation.
    - Naturally **penalizes constraint violations** while remaining negligible far from the constraint boundary, enabling **efficient exploration**.

3. **Comprehensive Experiments**
    - Conducted across **ten challenging tasks**, demonstrating the **consistency** of CSAC-LB in achieving **high returns** with **strict constraint satisfaction**.
    - Qualitative analysis of **exploration behavior** (e.g., SafeAntVelocity) shows that CSAC-LB can explore near constraint boundaries more effectively than baselines.
    - Visualization and analysis of **cost critic approximation errors** further support the method’s robustness and provide insight into architectural decisions.
---

### Weaknesses

1. **Lack of Explicit Theoretical Assumptions**
    - While the theoretical results are promising (e.g., suboptimality guarantee), **the assumptions are not clearly specified**. Does the guarantee hold for **arbitrary CMDPs**, or are there specific conditions required (e.g., bounded cost/reward, Lipschitz dynamics, compact state space)?
2. **Performance Trade-offs**
    - CSAC-LB is **not always the highest-performing method** in return, though it maintains safety. Discussion about **when and why CSAC-LB underperforms** in return (e.g., due to over-correction from the barrier function) could provide valuable insights and guide users.

---

Overall, this is an interesting and solid paper.

**Audience:**

Yes

**Audience Explanation:**

Safe RL would interest some researchers.

**Claims And Evidence:**

Yes

**Claims Explanation:**

They provided comprehensive experimental results.

**Requested Changes:**

Explicitly state the assumptions required for the theoretical guarantees, and clarify whether the results generalize to all CMDPs.

Discuss possible extensions, such as to multi-agent settings, time-varying constraints, or stochastic safety specifications.

---

> ### Author Response · Authors · 2025-08-19
>
> We thank Reviewer for the detailed and encouraging review, and for recognizing the strengths of our method and experiments. The reviewer's insightful suggestions will certainly help us improve the paper. We have revised the manuscript accordingly, with changes marked in blue.
>
> 1. On Lack of Explicit Theoretical Assumptions:
> > While the theoretical results are promising (e.g., suboptimality guarantee), the assumptions are not clearly specified... Does the guarantee hold for arbitrary CMDPs...?
>
> This is an excellent point, and we agree that a more formal discussion of the theoretical assumptions and their implications is needed.
> To answer the reviewer's question: our formal guarantees do not hold for arbitrary CMDPs. Our proof for Theorem 1 relies on two primary assumptions required by the theory of interior-point methods: 1. Strict Feasibility: The problem must have at least one policy π for which all constraints are strictly met (i.e., $J_C(\pi))<d$). 2. The objective $-J_R(\pi)$ and cost functions $J_C(\pi)$ must be convex. The proof for Theorem 2 relies on only the strict feasibility. As pointed out by the reviewer, we also require the assumption that the costs and rewards are bounded, as we want to analyse the suboptimality gap. Other assumptions (i.e., Lipschitz Dynamics and Compact State Space) are not required.
>
> In Section 4.4, preceding the theorems, we now explicitly state all formal assumptions required for the guarantees to hold (i.e., strictly feasible CMDP, convexity).
>
> 2. On Performance Trade-offs:
>
> > CSAC-LB is not always the highest-performing method in return, though it maintains safety. Discussion about when and why CSAC-LB underperforms in return... could provide valuable insights
>
> We thank the reviewer for this insightful question. A discussion of the performance comparison would indeed strengthen the paper. We investigated this exact behavior in our failure case analysis (Appendix A.3), where we discuss the performance drop in the SafetyPendulumTilt task. To make this analysis more prominent and to directly guide users, we have now dedicatedly pointed this out in the experiment section and completed with a guide on how to tune key hyperparameters in Appendix A.3.
>
> 3. On Possible Extensions:
> > Discuss possible extensions, such as to multi-agent settings, time-varying constraints, or stochastic safety specifications.
>
> These are very valuable suggestions for future work. We have incorporated these ideas into the future work part in the conclusion section. We briefly discuss how the principles of CSAC-LB could serve as a promising foundation for tackling these important and challenging domains.

---

### Review · Reviewer_MQEj · 2025-08-17

**Summary Of Contributions:**

The paper proposes The paper proposes an off-policy safe RL algorithm in constrained settings where its primary aim is to learn a policy that is recoverable from unsafe state transitions while being averse to unsafe states. The authors provide a comprehensive strategy to implement their intended goal by two mechanisms mainly (i) through the use of a smoothed log barrier function for policy recovery (ii) correcting underestimation / overestimation bias of the cost and reward  Q-values respectively so as to make the final policy to converge towards the true optimal policy. They demonstrate through varied experiments the effectiveness of their approach in terms of boosting both the performance and reducing incursions to unsafe states. They also provide certain desirable theoretical properties that emerge from employing the policy updates they propose.

Strengths:
1. I think the paper is written very well providing a good flow of thought while also providing the motivation pretty clearly.
2. Safety-constrained RL is a pressing challenge for real-world applications. The authors articulate well why existing methods (Lagrangian, CPO, clipping, etc.) are unstable or conservative.
3. I found the experimental setup convincing for the most part and found it to address the claims made in the paper.

Weaknesses:
1. My main concern is in the theoretical results. While understand and appreciate the sub-optimality guarantee,  I am concerned about this guarantee holding due to the restrictive convexity assumption on the $-J_r(\pi), J_c(\pi)$. Under the more general case of non-convexity, there exists no guarantee that the $-J_r(\pi*)$ closely matches the true intended objective $p*$. In such a case, can $\pi*$ indicate a policy that is minimally safe (much like SAC-Lag in the experiments). I want to emphasize that this is a theoretical nuance, but should be stated clearly in the paper. Can you provide your thoughts on this?
2. The paper suggests CSAC-LB is widely applicable without pretraining or prior knowledge, but all experiments are in simpler simulated benchmarks. I believe this is minor and may need to be discussed in a limitations section.

Questions:
1. When you mean, SAC penalty factor , I want to make sure you mena the following:|
Reward  = true_reward - penalty_factor * cost

2. Please add intuition for the suboptimality gap vs. barrier parameter trade-off in the main body, not just the appendix.

**Audience:**

Yes

**Audience Explanation:**

I believe the current work shows promise in the current safe RL landscape.

**Claims And Evidence:**

Yes

**Claims Explanation:**

The paper’s two core ideas are well supported: (i) the linear smoothed log barrier is precisely defined (Eq. 4.2) and contrasted against standard barriers with value/gradient plots showing stability and non-vanishing gradients in unsafe regions (Fig. 1), directly supporting the “stable recovery from violations” claim.

(ii) The pessimistic double cost-critic is motivated and ablated: using the max of two Qc’s counteracts cost underestimation; the ablation (varying the number of cost critics) shows a single critic predicts impossible negative costs and leads to violations, while two critics strike the best safety/return balance (Fig. 6).

**Requested Changes:**

Please try to address the weakness and questions in the summary of the review above.

---

> ### Author Response · Authors · 2025-08-22
>
> We thank the reviewer for the efforts and helpful suggestions, and the revised version is uploaded with new changes marked in blue.
>
> 1. On the Convexity Assumption in Theoretical Results:
>
> > My main concern is in the theoretical results. ... I am concerned about this guarantee holding due to the restrictive convexity assumption... Under the more general case of non-convexity, there exists no guarantee... Can you provide your thoughts on this?
>
> This is a crucial point, and we agree completely. The convexity assumption is a theoretical simplification that does not hold for deep reinforcement learning, and our intention was for the theory to provide valuable intuition for the algorithm's design.
>
> In practice, CSAC-LB's ability to learn safe and high-performing policies stems from its core algorithmic mechanisms. The smoothed log barrier provides a stable, non-vanishing gradient that allows for effective recovery from constraint violations, while the pessimistic critic architecture robustly guards against cost underestimation. These mechanisms ensure practical safety even when the formal theoretical guarantees do not apply.
>
> It is worth noting a practical consideration for scenarios with extreme reward-cost imbalances. If the magnitude of the reward signal significantly outweighs the penalty from the cost function, the policy may prioritize reward to the detriment of safety. This can be effectively mitigated using standard techniques such as reward normalization or by increasing the log barrier factor $\mu$.
>
> We have added a discussion immediately following the theorems in Section 4.4, where we clarify that while the guarantees may not hold formally, the theory provides intuition for the algorithm's behavior. Our empirical results (Fig. 3) show the exact behavior in practice when the assumptions are violated. We will also add this to the newly added "Limitations" section.
>
>
>
> 2. On Applicability and Simulated Benchmarks:
>
> > The paper suggests CSAC-LB is widely applicable without pretraining or prior knowledge, but all experiments are in simpler simulated benchmarks. I believe this is minor and may need to be discussed in a limitations section.
>
> We agree. To provide a more balanced perspective, in the Limitations section, we acknowledge that our experiments were conducted exclusively in simulation and have also added this as a potential direction for future work.
>
> 3. Questions:
>
> 3.1
> > When you mean, SAC penalty factor, I want to make sure you mean the following: Reward = true_reward - penalty_factor * cost
>
> Yes, that is precisely how we implemented the reward shaping baseline. We have added a sentence to Section 5.1 to clarify this for the reader.
>
> 3.2
> > Please add intuition for the suboptimality gap vs. barrier parameter trade-off in the main body, not just the appendix.
>
> Thank you for this excellent suggestion. We agree it improves the paper's flow. We have moved this discussion into the theory-practice gap part in Section 4.4, as it fits perfectly there.

---

> > ### Comment · Reviewer_MQEj · 2025-09-18
> >
> > Thank you for addressing these concerns in a methodical fashion. It addresses all the questions and concerns I had.

---

> ### Comment · Reviewer_MQEj · 2025-09-10
>
> Could the authors respond to the questions I raised.
>
> Thanks

---

### Review · Reviewer_uWry · 2025-08-26

**Summary Of Contributions:**

The paper proposes CSAC-LB, a variant of Soft Actor-Critic (SAC) that incorporates a linear smoothed log barrier function and a pessimistic double-critic design for handling constraints. The goal is to improve training stability and safety in constrained reinforcement learning (RL). The paper provides theoretical analysis and experimental evaluation across multiple benchmark tasks.

**Audience:**

Yes

**Audience Explanation:**

The paper proposes an interesting and practically relevant method with clear theoretical and empirical contributions.

**Claims And Evidence:**

Yes

**Claims Explanation:**

The paper addresses an important and timely problem in constrained RL, where balancing performance with safety is critical.

The idea of integrating a smoothed log barrier function into the SAC framework is novel and provides a stable alternative to standard methods.

The use of pessimistic double critics for costs is well-motivated and supported by ablation experiments.
The paper is clearly written, and the proposed method is presented in a structured manner.

**Requested Changes:**

1.	Insufficient references in Introduction:

The introduction makes several strong claims about the limitations of existing constrained RL methods (e.g., instability, conservatism, difficulty in handling constraints), but it lacks sufficient recent references to back these points. The authors should expand the literature coverage, particularly to include recent advances.

2.	Baselines are outdated:

The experimental section primarily compares CSAC-LB against classical baselines such as SAC, SAC-Lag, WCSAC, and CPO before 2021. You can compare with recent safe RL methods to strengthen the empirical evaluation.

3.	Performance vs. SAC-Lag:

From the reported results, it appears that SAC-Lag sometimes achieves higher returns. Since SAC-Lag is still a widely used baseline, a deeper analysis of why CSAC-LB underperforms in return compared to SAC-Lag in certain cases would be useful. For example, is the trade-off between safety and return inherent to the barrier design, or is it a matter of hyperparameter tuning?

---

> ### Author Response · Authors · 2025-08-29
>
> We thank the reviewer for their valuable suggestions and constructive feedback. A revised version of our manuscript has been uploaded with the changes marked in blue.
>
> 1. On Insufficient References in the Introduction
> > Insufficient references in Introduction: The introduction makes several strong claims about the limitations of existing constrained RL methods (e.g., instability, conservatism, difficulty in handling constraints), ... include recent advances.
>
> We agree that including more recent literature is essential for accurately framing the challenges in the current constrained RL landscape. We have revised the introduction (Section 1) to incorporate several recent citations (including from 2022, 2024, and 2025) to further support our claims. These additions specifically support our claims regarding the limiting assumptions of some existing methods and the common trade-off between "reckless" and "conservative" policies. We believe these changes better situate our contributions within the literature and strengthen the motivation for our approach.
>
> 2. On Including More Recent Baselines
> > Baselines are outdated: The experimental section primarily compares CSAC-LB against classical baselines such as SAC, SAC-Lag, WCSAC, and CPO before 2021... the empirical evaluation.
>
> We agree that comparing against a more recent baseline provides a valuable perspective on our method's performance. Following the reviewer's advice, we have implemented and added APPO [1] to our experimental comparison.
>
> In our experiments, we found that using the hyperparameters suggested in the official codebase, APPO exhibits highly conservative behavior. While it successfully adheres to safety constraints, its episodic returns were notably lower than our method's and even the CPO baseline's. Despite our best efforts, we were unable to reproduce the performance levels reported in the original publication on our set of tasks.
>
> We have included these new results and our implementation code in the updated figures and supplementary materials for full transparency. The expanded comparison continues to show that our method, CSAC-LB, demonstrates a superior balance of achieving high returns while strictly satisfying safety constraints.
>
> 3. On the Performance Trade-off vs. SAC-Lagrangian
> > Performance vs. SAC-Lag: From the reported results, it appears that SAC-Lag sometimes achieves higher returns. Since SAC-Lag is still a widely used baseline, ... For example, is the trade-off between safety and return inherent to the barrier design, or is it a matter of hyperparameter tuning?
>
> The performance difference between CSAC-LB and SAC-Lag is indeed an inherent and intentional trade-off stemming from our barrier design, which deliberately prioritizes robust safety.
>
> SAC-Lagrangian's multiplier is reactive—it only increases after violations occur. This can lead to a policy that aggressively pursues high rewards and only gets penalized later, resulting in high returns but fundamentally unsafe behavior, as shown in our experiments (e.g., Fig. 3 and 4). In contrast, our log barrier is proactive, acting as a static, preventative "repulsive force" from the constraint boundary at all times. This design choice prevents the agent from exploring deeply into unsafe regions in the first place, leading to a slightly more conservative but far more reliable and safer policy. Our analysis (e.g., Fig. 7) confirms that CSAC-LB consistently converges to the safe, high-performing region of the return-cost space when tuning the hyperparameters of the log barrier function.
>
> [1] Juntao Dai, Jiaming Ji, Long Yang, Qian Zheng, Gang Pan: Augmented Proximal Policy Optimization for Safe Reinforcement Learning. AAAI 2023: 7288-7295

---

### Decision · Action_Editor_xRT3 · 2025-10-13

**Recommendation:** Accept as is

**Audience:**

Yes

**Audience Explanation:**

Safe RL is an active area of research and is critical for many applications. Constrained MDPs are currently the de-facto standard in safe RL. The paper introduces some novel ideas and positions them well within the related literature.

**Claims And Evidence:**

Yes

**Claims Explanation:**

The authors propose a safe RL method for constrained MDPs based on SAC, log barrier regularization, and a pessimistic double critic.
Theoretical results hold under unrealistic assumptions (namely convexity) but are nonetheless interesting.
Experiments show the performance, stability, and safety of the proposed algorithm comparing with relevant baselines.
The main issues raised by the reviewers are:
- Limited discussion of recent related works
- Outdated baselines for experiments
- Theory assumptions not sufficiently discussed
However, these were all fixed during the discussion phase by adding more references, discussions and experiments.
So the paper can be published as is.